# Pharmacovigilance in Brazil: The Government Monitoring of Adverse Events Reported from COVID-19 Vaccine—A Narrative Review

**DOI:** 10.3390/healthcare12030371

**Published:** 2024-02-01

**Authors:** Mariana Carvalho de Moraes, Ivone Duarte, Rui Nunes

**Affiliations:** Department of Social Sciences and Health, Faculty of Medicine, Universidad do Porto, 4200-319 Porto, Portugalruinunes@med.up.pt (R.N.)

**Keywords:** pharmacovigilance in vaccines, adverse events, vaccines in Brazil, COVID-19 in Brazil

## Abstract

Background: Is pharmacovigilance at a moment of prominence for science, and in relation to governments’ responsibilities towards their nations, as the new coronavirus pandemic has surprised everyone in a negative and lethal way? Objective: Evaluate pharmacovigilance as a resource for controlling and understanding adverse events caused by vaccines in use. Methods: This is a narrative review of the literature. Scientific articles available in databases, government bulletins and similar bodies were used. The search was carried out using the descriptors: “Pharmacovigilance AND COVID-19 in Brazil”, “Vaccine Development AND COVID-19”, “Vaccination Hesitancy AND COVID-19”, “Public Health Surveillance AND COVID-19”. The period from May 2021 to June 2022 was covered. Results: The occurrence of some adverse events was observed, including cases of allergy, myocarditis and rheumatoid arthritis. It is important to highlight that these adverse events were identified as rare, occurring in a small percentage of the vaccinated population. Despite these adverse events, the benefits of vaccines proved to be essential for controlling the pandemic. Conclusions: The information presented highlights the importance of pharmacovigilance to continuously monitor and evaluate the safety of vaccines, identifying any potential adverse events early. This balance between risk and benefit emphasizes the need for a careful and informed approach when making decisions about vaccination policies, prioritizing public health and population safety.

## 1. Introduction

Pharmacovigilance is defined as the science that aims to monitor the safety of pharmaceutical products, including drugs and vaccines, and takes measures to reduce their risks and increase their benefits [1]. Thus, it aims to monitor the risk–benefit ratio, as well as improve patient safety and quality of life [1,2].

The onset of the new coronavirus disease 2019 (COVID-19) surprised the world in an unprecedented and negative way, exposing human frailty [3,4]. The work of the global scientific community stunned the world for its efficiency and effectiveness when mapping the virus genome, as well as the record-breaking time the vaccines were created in due to the urgency imposed by the aggressiveness of the virus. The speed of vaccine development also frightened global populations [5,6], leading people to mistrust them and reject offers of immunization [7].

In Brazil, four vaccines were approved to be used without restrictions for people over eighteen years old. Cases of adverse events were emerging and concerning the scientific community, leading managers to create pharmacovigilance protocols in order to map and minimize the most severe collateral damage.

Pharmacovigilance supports science and helps citizens who have doubts about the use of the vaccine, which are exacerbated by misinformation. This can spread rapidly thanks to the widespread use of the internet. The objective of this study was to evaluate government pharmacovigilance actions as a resource for controlling and understanding the adverse events reported from COVID-19 vaccine use in Brazil.

## 2. Methods

The present study is a narrative literature review carried out by analyzing documents, newsletters from government websites and articles. The databases used to search for articles were PubMed, MedScape, Scielo and government websites of ANVISA and the Ministry of Health. The following descriptors were used for the selections identified in Medical Subject Headings and Health Sciences Descriptors: “Pharmacovigilance AND COVID-19 AND Brazil”, “Vaccine Development AND COVID-19 AND Brazil”, “Vaccination Hesitancy AND COVID-19 AND Brazil”, “Pharmacovigilance AND COVID-19”, “Vaccine Development AND COVID-19”, “Vaccination Hesitancy AND COVID-19”, Pharmacovigilance AND COVID-19, Vaccine Development AND COVID-19, Vaccination Hesitancy AND COVID-19, “Public Health Surveillance AND COVID-19”. The research was conducted between May 2021 and June 2022.

Duplicates and incomplete studies that did not have the necessary information to address the topics proposed in the study were excluded. Subsequently, a critical analysis was performed, and the data were collected using an Excel spreadsheet with the following information: title, year of publication and authors. Then the content was analyzed and used for the development of the study.

## 3. Results and Discussion

From the database search, three journal articles, four government bulletins and 535 scientific articles were identified (Table 1). Of the total 542 bibliographic materials, 36 were selected according to the selection criteria and objectives of the study, as shown in Figure 1.

### 3.1. Pharmacovigilance: From the Fundamental Pillars to the Challenges of Monitoring throughout History

The concept of pharmacovigilance established by the World Health Organization (WHO) has four main pillars related to adverse events and problems related to the use of medicines: identification, evaluation, understanding and prevention.

When administering a drug or study drug, in addition to the useful therapeutic effects, certain undesirable events occur in some people. Researchers are emphatic in reporting that there is no medication without the risk of adverse events. The probability of occurrence may vary, the reaction may be mild or severe, it may be predictable or not, but the doctor/researcher and the patient/research subject must always be aware of the possibility of its appearance [8].

According to the WHO, pharmacovigilance is defined as “the science and activities related to the identification, evaluation, understanding and prevention of adverse events or any other problems related to the use of medicines and vaccines” [9].

Analyzing the temporal evolution, we found records of incidents related to the use of medicines from the end of the 19th century. There were cases such as sudden death attributed to the use of chloroform in anesthesia, and jaundice resulting from the use of arsenic in the treatment of syphilis [10]. However, a crucial point in the history of global pharmacovigilance was the thalidomide disaster of the 1950s and 1960s. During this period, thousands of cases of phocomelia, a rare congenital malformation, were associated with the use of thalidomide to treat or prevent nausea in pregnant women. The impact was significant, with an increase from 1.5% to 20% in the incidence of congenital malformations in women who ingested thalidomide during pregnancy. It is important to highlight that a review of studies carried out in the pre-commercial phase indicated that the data had been misinterpreted [2,11,12].

After the global tragedy caused by thalidomide in 1961, there was a need for international efforts to address safety issues relating to medicines. The 16th World Health Event (1963) adopted a resolution reaffirming the need for immediate action regarding the rapid dissemination of information on adverse drug reactions. 

From 1962 onwards, the North American regulatory agency (Food and Drug Administration) began to demand more rigorous non-clinical and clinical studies from drug manufacturers. In 1968, the WHO began the pilot phase of the International Program for Monitoring Adverse Drug Reactions, in which 10 countries were involved. This led to the creation of the International Drug Monitoring Program, which since 1978 has been implemented by the Uppsala Monitoring Center (UMC) in Sweden. Today, the program brings together more than 140 countries in the WHO’s worldwide pharmacovigilance network [13].

### 3.2. Healthcare in Brazil: ANVISA and Advanced Pharmacovigilance Strategies

The Brazilian government, through the National Health Surveillance Agency (ANVISA), exercises control over the use of medicines, based on RCB n° 406/2020, regarding Good Pharmacovigilance Practices for Registrars of Medicines for Human Use [14]. For the adequate management of adverse events post-vaccination (AEPV) associated with a new vaccine, ANVISA highlights the importance of a sensitive surveillance system to assess the safety of the product and promptly respond to the population’s concerns. These activities include early notification and investigation of events, forming part of an AEPV Surveillance cycle. This cycle encompasses the detection of suspected cases, notification, registration in an information system, detailed investigation (involving clinical and laboratory tests, among others), active searches for new events, evaluation of information, classification of causality and timely feedback.

As a result of the COVID-19 pandemic, vaccines were produced in record time. Vaccines already in circulation also cause adverse events, as does any drug in use. The Health Units that administer immunobiological medicines (vaccines, serums and immunoglobulins) must notify and investigate these occurrences and register them in the PNI Information System-Post-Vaccination Adverse Event (SIPNI-EAPV) in order that they can be analyzed by the state and at national level [15].

Since the beginning of the distribution of the COVID vaccines in Brazil in early 2021, the population was already concerned about their potential adverse events. Even so, Henze [16] argues that “the observed effects of the vaccine last little and the benefits of the vaccine outweigh”.

Adverse events from COVID-19 vaccines identified by health services are reported online by healthcare professionals through the e-SUS system (https://notifica.saude.gov.br/, accessed on 24 April 2022). These records undergo investigation and conclusion by the post-adverse event surveillance of the Municipal and State Immunization Coordination, with subsequent review and support from the Ministry of Health. Given the nature of the disease, which is constantly being researched due to its evolving variants, this cycle is repeated continuously, and it is crucial that healthcare professionals and citizens notify official bodies about any adverse events [17,18].

Vigimed, a tool established in agreement between ANVISA and UMC, the WHO collaborating center, is used as pharmacovigilance software for international medicines monitoring. After recording adverse events in Vigimed, reports are evaluated for severity, associated risk, predictability and causal link between the adverse event and the medicine or vaccine. Depending on the case, the notification may result in an investigation and various measures, including communication of health risks, dissemination of warnings, changes to leaflets, restrictions on use, request for batches and even the cancellation of the medicine or vaccine’s health registration.

### 3.3. Benefits, Challenges and Adverse Events Reported from COVID-19 Vaccines in Brazil and around the World

The high morbidity and mortality rate of the COVID-19 pandemic has led to the larg-est and most diverse effort in vaccine development in history. Approximately 7 months after characterizing the severe acute respiratory syndrome coronavirus 2 (SARS-CoV-2) viral genome, around 200 vaccines with different production platforms were at different stages of development, including at least five in phase 3 trials. Although no vaccine can guarantee absolute protection, it is undeniable that immunization played a crucial role in preventing deaths and serious cases of COVID-19, contributing significantly to containing the spread of the pandemic [19]. In Brazil, according to research carried out in Londrina, Paraná, it showed that 75% of deaths from COVID-19 recorded in the first 10 months of 2021 occurred in individuals who were not immunized against the disease [20]. Unvaccinated elderly people died at almost three times the rate of those who were immunized. Among people under 60 years of age, the number of deaths among unvaccinated individuals was three times higher than among immunized individuals [21,22].

Vaccination against COVID-19 in Brazil began in the second half of January 2021 with two vaccines from different laboratories: ChAdOx1 nCoV-19 developed by the British pharmaceutical group AstraZeneca, in partnership with the University of Oxford and technology transfer to the Oswaldo Foundation Cruz (FIOCRUZ) in Rio de Janeiro, and CoronaVac, a Butantan vaccine produced in partnership with the Chinese biopharmaceutical company Sinovac. In May, a third BNT162b2 mRNA COVID-19 vaccine (Pfizer–BioNTech, New York, NY, USA) and a fourth Ad26.COV2.S vaccine were added. The Ad26.COV2.S vaccine (Janssen/Johnson & Johnson, Titusville, NJ, USA) was acquired for use in the Brazilian territory in June 2021. Since the beginning of the vaccination program, all four of these vaccines have been authorized by ANVISA for use in Brazil, and have been widely offered by the National Immunization Program (PNI) of the Ministry of Health.

Rare adverse events have been described following immunization with approved COVID-19 vaccines. In April 2021, five patients had venous thrombosis and thrombocytopenia after receiving the first dose of the adenoviral vector vaccine ChAdOx1nCoV-19. These five cases occurred in a population of over 130,000 vaccinated people, thus representing a rare condition defined as vaccine-induced immune thrombotic thrombocytopenia [23,24,25].

In the United States, the Ad26.COV2.S COVID-19 vaccine (Janssen/Johnson & Johnson) was approved for emergency use on 27 February 2021. As of 12 April 2021, approximately 7 million doses of the Ad26.COV2.S vaccine had been administered in the US, and six cases of cerebral venous sinus thrombosis with thrombocytopenia were identified among recipients, resulting in a temporary national pause in vaccination with this product on 13 April 2021 [26,27,28].

Immediately after authorization was granted in the United Kingdom (UK) and the United States (US) in early/mid-December 2020, there were unique reports of reactions from hypersensitivity in a very small number of patients, possibly due to a component in the Pfizer–BioNTech BNT162B2 or Moderna (Cambridge, MA, USA) mRNA-1273 vaccines. There is a consensus that these vaccines are contraindicated only when there are allergies to preparations in similar new mRNA technologies. These reactions, which were resolved after treatment, were caused by either an active component of the vaccine or severe allergic reaction to the first dose [29]. Rare cases of myocarditis and pericarditis have been reported, especially in male adolescents and young adults several days after COVID-19 mRNA vaccination (Pfizer–BioNTech or Moderna) [30].

The CoronaVac vaccine in phase 1 and 2 trials showed good safety, tolerability and immunogenicity in healthy adults 18 years of age and older. However, there are reports of pityriasis rosea that developed 4 days after the first dose of the vaccine and were present for 1 week in a phase 3 clinical trial conducted in Turkey [31]. Acute rheumatoid arthritis has also been reported for 18 days with the presence of swelling and pain in the left knee joint after CoronaVac vaccination.

According to the International Council for Harmonization of Technical Requirements for Pharmaceuticals, the systematic observation of adverse events caused by immunobiological medicinal products marketed by established practices is essential for periodic risk–benefit assessment in comparison with known adverse events, as well as for the awareness of rare adverse events not described in drug leaflets [32].

ANVISA, considering international specifications, performed its control of EAPV considering the Organ Class System and Term Preference, and calculated its incidence per 1000 doses applied to non-serious events, and 100,000 doses applied to severe and rare events [33].

In periodic bulletins, the agency reports that very rare, serious adverse events and deaths are still discussed weekly in the Inter-institutional Committee for Pharmacovigilance of Vaccines—formed by the PNI/SVS, Pharmacovigilance Management, National Health Surveillance Agency, National Institute of Health Quality Control, in addition to specialists with expertise in vaccinations and pharmacovigilance of vaccines, including immunologists, infectiologists, neurologists, cardiologists, rheumatologists and pediatricians.

In the first four months of the campaign (18 January to 23 May 2021), 74,563 cases of EAPV were reported. The incidence of EAPV is observed per 1000 doses administered each day of vaccination. Of the reported EAPV, 70,110 were classified as EANG, and 4453 events were classified as EAG, of which 2277 were related to deaths [34]. Table 2 shows the incidence of EAPV by type of vaccine, corresponding to the first five months of the campaign, still without the use of Janssen’s immunization.

It is noteworthy, however, according to information provided by the Ministry of Health, that when comparing the incidences of adverse events between different vaccines, it is crucial to consider the proportion of the population immunized with each vaccine. It must be highlighted that vaccination was initially targeted at the most vulnerable groups, such as the elderly, who are more prone to experiencing serious adverse events (which could be events related to other conditions and not necessarily the vaccines), and healthcare professionals, who are more likely to report adverse events. Furthermore, it is important to note that the incorporation of the different vaccines occurred sequentially. Thus, as the data presented indicate, pharmacological surveillance has been diligently conducted, adjusting to vaccination cycles as different groups become eligible for immunization [33]. This approach aims to ensure a comprehensive and contextualized analysis of vaccine safety, considering the specific characteristics of each target population throughout the immunization process.

The systematic observation of adverse events caused by immunobiological medicinal products marketed by established practices is essential for the periodic evaluation of the risk–benefit ratio compared with known adverse events, as well as for the knowledge of rare and unspecified adverse events [21].

It is worth remembering that Brazil is internationally recognized for its National Vaccination Plan. Created in 1973, the program makes vaccines available to the population through Brazil’s Unified Health System, one of the largest free health-service delivery programs in the world. Through widescale vaccination, Brazil has already eradicated diseases such as smallpox and polio (a cause of childhood paralysis), offering all vaccines recommended by the World Health Organization to its population [29].

During the year 2020, the race for the creation and effectiveness of the COVID-19 vaccination raised ethical debates that continue to be discussed, which implies the acceptance of the population to take the vaccines offered in Brazil, especially when one thinks about their adverse events.

Colgrove [35] said that, among the risks related to vaccines, non-vaccination is considered the most significant. The deleterious effects associated with the use of vaccines, when scientifically proven, occur very rarely, and are insignificant when compared with the risks related to non-vaccination.

Strategies to encourage the use of vaccines, the author continues, “are traditionally adopted in public health, but may be insufficient to ensure an increase in vaccine coverage” [35]. In this context, it is necessary to maintain a clear understanding of the value of vaccines.

Both in the general population and among healthcare professionals, there are some who demonstrate resistance to immunization in all parts of the world, either due to their personal beliefs, or because they are influenced by misinformation widely disseminated online.

Vaccination continues to be vital, as it helps reduce the proliferation of a disease. The mutating virus can still infect those who are vaccinated, and vaccination does not interrupt the disease cycle.

Corroborating this statement, Carbinatto [36] observes that different sequences indicate that SARS-CoV-2 mutates, forming “sub-groups” of the same virus. According to estimates, the study points out, this has been happening at a frequency of about one mutation per month. But these changes also reveal which viruses are closest to each other and which are more distant, by allowing the construction of a “family tree”.

Variants of the disease have already been circulating worldwide and, in several countries with high levels of vaccinations, restrictive measures have again been imposed, with a view to reducing the infection rates, including in those who have received the available vaccination. This is due to more aggressive and lethal variants that are still being studied in order to adapt vaccinations. Adverse events are common to any medicine; they are frightening due to the unprecedented nature of COVID-19, but they do not discredit the effectiveness of vaccination.

## 4. Conclusions

Although the urgency for developing effective and safe vaccines is crucial in pandemic scenarios, clinical development for emergency-use authorization and licensing is a considerable challenge. In this context, pharmacovigilance of vaccine safety and surveillance of virus variants are essential to ensuring the wellbeing of the population.

The Brazilian data for adverse events reported from the COVID-19 vaccine, although the country is continental in its size, are acceptable given the number of deaths recorded. It is known that this investigative process must be a permanent fixture. It requires the cooperation of the network involved in combating coronavirus, which includes newly immunized groups, newly approved immunizers acquired by governments, and healthcare professionals working for the immunization program.

Pharmacovigilance is effective and needs, as is the case for many other public services, to be widely understood to raise awareness in populations that at times disregard such salutary and indispensable information in times of misinformation and resistance to free medical treatments available in Brazil.

Lastly, it is necessary to credit science and trust in public measures to control this disease and all others that require the use of drugs, because the agency responsible for health control is committed to defeating them, there is transparency in the policies used, and the key information is widely available. The Brazilian population can credit the Unified Health System and its assistance programs, with emphasis on the National Vaccination Program. The temporal constraint in the study limits the immediate analysis of acute adverse events reported from the COVID-19 vaccine. Nevertheless, it sets the stage for future investigations, highlighting the opportunity to assess the mid- to long-term events of these vaccines.

## Figures and Tables

**Figure 1 healthcare-12-00371-f001:**
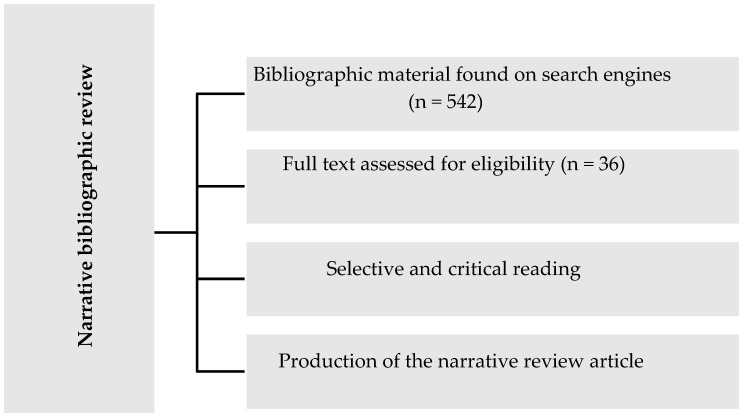
Flow diagram of the study selection process.

**Table 1 healthcare-12-00371-t001:** Details of the search strategy.

Database	Search Strategies	Papers Found
Pubmed	“Pharmacovigilance AND COVID-19 in Brazil”	17
Pubmed	“Vaccine Development AND COVID-19 in Brazil”	234
Pubmed	“Vaccination Hesitancy AND COVID-19 in Brazil”	32
Pubmed	“Public Health Surveillance AND COVID-19”	220
MedScape	“Pharmacovigilance AND COVID-19”	1
MedScape	“Vaccine Development AND COVID-19”	4
MedScape	“Vaccination Hesitancy AND COVID-19”	2
Scielo	Pharmacovigilance AND COVID-19	3
Scielo	Vaccine Development AND COVID-19	20
Scielo	Vaccination Hesitancy AND COVID-19	9

Pubmed = A free resource developed and maintained by the National Library of Medicine (NLM^®^) in the United States; MedScape = The leading online destination for physicians and healthcare professionals worldwide, providing the latest medical news and expert perspectives, essential information about medications and diseases, as well as continuing medical education content; Scielo = Scientific Electronic Library Online. COVID-19 = coronavirus disease 2019.

**Table 2 healthcare-12-00371-t002:** Cumulative incidence of adverse events after coronavirus disease 2019 vaccination according to severity, vaccine and overall incidence (per 100,000 doses administered), January–May, Brazil, 2021.

	AstraZeneca/Fiocruz		Sinovac/Butantan		Pfizer/Wyeth		Total
EAPV	No.	Incidence	No.	Incidence	No.	Incidence	Overall Incidence
Severe	1397	9.6	3026	10.3	30	3.9	10.0
Death	556	3.8	1717	10.8	4	0.5	5.1
Non-serious	43,489	299.5	26,234	89.2	387	49.9	156.8

EAPV = postvaccination adverse event. Research Source: Brazil’s National Immunization Program.

## Data Availability

No new data was created or analyzed in this study. Data sharing is not applicable to this article

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
