# Peer review of "Pharmacovigilance in Brazil: The Government Monitoring of Adverse Events Reported from COVID-19 Vaccine—A Narrative Review"

_healthcare, 2024, doi:10.3390/healthcare12030371_

Round 1

Reviewer 1 Report

Comments and Suggestions for Authors

Comments on the Quality of English Language

Author Response

Dear Reviewer,

I would like to express my sincere gratitude for the time and effort dedicated to reviewing the scientific article, entitled "PHARMACOVIGILANCE IN BRAZIL: THE GOVERNMENT MONITORING OF THE ADVERSE SIDE EFFECTS CAUSED BY VACCINES AGAINST COVID-19 – A NARRATIVE RE-VIEW". His expertise and insight were invaluable in improving the quality and clarity of the work.

His detailed observations and constructive suggestions demonstrated a deep understanding of the subject, significantly enriching the content. I am especially grateful for your patience and dedication in pointing out specific aspects that required attention, which undoubtedly contributed to raising the quality of the article as a whole.

Furthermore, your quick response and readiness to provide feedback were essential in meeting the established deadlines. Your attention to detail and commitment to academic excellence have not gone unnoticed and are truly appreciated.

1- Abstract
Response: The study design item was unified with the method session. The results and conclusion items have been reformulated.

2- Introduction
Response: The second paragraph was reformulated as suggested.

Methods
Response 1: Government websites were included in the search locations.
Response 2: Including terms such as "side effects", "adverse events", "adverse effects" and "unwanted effects" in Covid-19 vaccine research would be a more comprehensive practice as it would help capture a wider range of related information possible adverse reactions, contributing to a more complete assessment of the safety of the immunizer. The specific non-inclusion of these terms aims to choose more technical and standardized keywords to ensure consistency in data collection. Furthermore, we opted for more formalized terms that comply with the standards established in the scientific literature, ensuring precision in the analysis of results. This approach seeks to maintain the integrity and reliability of the data obtained.

Results, general comment and recommendation:
Response: The movement of text in the results and discussion was carried out as suggested, however, we find your recommendation to merge results and discussion pertinent. Your detailed observations and request for more clarity and avoidance of ambiguity were extremely valuable and impactful. Through
  It was possible to improve the quality of the article through its guidance and insights into the text, making it more accessible and understandable for the target audience.

Once again, thank you for all your contributions!
Yours sincerely,
Mariana Carvalho

Reviewer 2 Report

Comments and Suggestions for Authors

References should be provided and improved in the background. A lot of sentences have no reference. Eg. first phrase. 

Please explain further the MeSH terms used and descriptors. There are not information duplicated?

Search was performed between May 2021 and 2022. Covid, however started in 2020. Vaccines started in December 2020. This is a limitation. Should be discussed. 

Why the details in search strategy are different among platforms of search?

Why LILACS information was not provided in the table 1?

Flow diagram have information missing? What type of diagram is that? Some simplified representation of PRISMA?

3.1 Section information should be in the introduction. It seems to me a lot of history, not relevant for the study provided. 

Your objective in the title is side effects to COVID VACCINES IN BRAZIL, in the ABSTRACT is adverse effects by vaccines (in Brazil???). And the results seems to be generally in other way. It all should have the same focus. 

Additionally, the terms side effects, adverse effects and adverse reactions are used interchangely. Please choose the one is more related to your study and use consistently. 

References should be provided and improved in the background. A lot of sentences have no reference. Eg. first phrase. 

Please explain further the MeSH terms used and descriptors. There are not information duplicated?

Search was performed between May 2021 and 2022. Covid, however started in 2020. Vaccines started in December 2020. This is a limitation. Should be discussed. 

Why the details in search strategy are different among platforms of search?

Why LILACS information was not provided in the table 1?

Flow diagram have information missing? What type of diagram is that? Some simplified representation of PRISMA?

3.1 Section information should be in the introduction. It seems to me a lot of history, not relevant for the study provided. 

Your objective in the title is side effects to COVID VACCINES IN BRAZIL, in the ABSTRACT is adverse effects by vaccines (in Brazil???). And the results seems to be generally in other way. It all should have the same focus. 

Additionally, the terms side effects, adverse effects and adverse reactions are used interchangely. Please choose the one is more related to your study and use consistently. 

English should be improved. Some wording in Portuguese is present in the manuscript. Eg. Page 7 after table. 

References should be confirmed: Eg: "Turkey33", Reference 36 is missing?

References should be confirmed: Eg: "Turkey33", Reference 36 is missing?

Comments on the Quality of English Language

English should be improved. Some wording in Portuguese is present in the manuscript. Eg. Page 7 after table. 

Author Response

Dear Reviewer,

I would like to express my sincere gratitude for the time and effort dedicated to reviewing the scientific article, entitled "PHARMACOVIGILANCE IN BRAZIL: THE GOVERNMENT MONITORING OF THE ADVERSE SIDE EFFECTS CAUSED BY VACCINES AGAINST COVID-19 – A NARRATIVE RE-VIEW". His expertise and insight were invaluable in improving the quality and clarity of the work.

His detailed observations and constructive suggestions demonstrated a deep understanding of the subject, significantly enriching the content. I am especially grateful for your patience and dedication in pointing out specific aspects that required attention, which undoubtedly contributed to raising the quality of the article as a whole.

Furthermore, your quick response and readiness to provide feedback were essential in meeting the established deadlines. Your attention to detail and commitment to academic excellence have not gone unnoticed and are truly appreciated.

  1. References should be provided and improved in the background. A lot of sentences have no reference. Eg. first phrase.

Pharmacovigilance is defined as the science that aims to monitor the safety and take measures to reduce the risks and increase the benefits of pharmaceutical products, in-cluding drugs and vaccines1. Thus, its actions are aimed at monitoring the risk/benefit ratio, as well as improving patient safety and quality of life.1,2

  1. Please explain further the MeSH terms used and descriptors. There are not information duplicated?

The best descriptors were selected using Medical Subject Headings (MESH - https://www.ncbi.nlm.nih.gov/mesh/): KEYWORDS: Pharmacovigilance in vaccines; Adverse effects to vaccines; Vaccines in Brazil; COVID-19 in Brazil. Duplicates were removed.

  1. Search was performed between May 2021 and 2022. Covid, however started in 2020. Vaccines started in December 2020. This is a limitation. Should be discussed.

Certainly, it is relevant to discuss the temporal limitation of the study concerning the acute adverse effects of COVID-19 vaccines. The data collection confined between May 2021 and 2022 may constrain the comprehensive analysis of immediate adverse effects of the vaccines, given that the vaccination commenced in December 2020.

However, it is crucial to emphasize that this temporal restriction does not negate the significance of the study. Although the research has a limited scope in assessing the vaccines' acute effects, it can provide a robust foundation for exploring mid- to long-term effects. This limitation, in fact, may pave the way for future longitudinal investigations, enabling a more comprehensive and detailed analysis of potential adverse effects over a broader timeframe.

Thus, while the temporal constraint may limit the immediate examination of acute adverse effects of the vaccines, it sets a promising stage for ongoing research, facilitating broader inquiries into the mid- and long-term impacts of COVID-19 vaccines.

  1. Why the details in search strategy are different among platforms of search?

The search in the different databases was carried out according to the strategy described in table 1.

  1. Why LILACS information was not provided in the table 1?

In the search execution phase, the LILACS database was not used. The manuscript methodology has been adjusted.

  1. Flow diagram have information missing? What type of diagram is that? Some simplified representation of PRISMA?

The flow diagram was created to illustrate the data selection process and was not inspired by the Preferred Reporting Items for Systematic Reviews and Meta-Analyses (PRISMA) diagram, as the present study is a narrative review. Since PRISMA is an evidence-based minimum set of items for reporting in systematic reviews and meta-analyses, it is not applied to the present study.

  1. 1 Section information should be in the introduction. It seems to me a lot of history, not relevant for the study provided.

According to detailed guidance from another reviewer, the results and discussion topics were restructured for greater clarity and to avoid ambiguities. We believe that with this new version the article is more accessible and understandable for the target audience.

  1. Your objective in the title is side effects to COVID VACCINES IN BRAZIL, in the ABSTRACT is adverse effects by vaccines (in Brazil???). And the results seems to be generally in other way. It all should have the same focus.

Considering the review, the following modifications were made:

  • The title of the manuscript has been adjusted to PHARMACOVIGILANCE IN BRAZIL: THE GOVERNMENT MONITORING OF THE ADVERSE SIDE EFFECTS CAUSED BY VACCINES AGAINST COVID-19 – A NARRATIVE RE-VIEW
  • Section 3.3 has been changed to Adverse effects: 3.3. Adverse effects

  1. Additionally, the terms side effects, adverse effects and adverse reactions are used interchangely. Please choose the one is more related to your study and use consistently.

To better align the objectives, results and discussion of the research, the term adverse effects and adverse side effects were used.

  1. References should be provided and improved in the background. A lot of sentences have no reference. Eg. first phrase.

The limitation of references occurs due to the period during which the study was carried out, at that time there was a shortage of scientific articles, especially in Brazil as the study refers to an immediate analysis of the acute adverse effects of COVID-19 vaccines. However, our article reinforces the importance of future investigations, highlighting the opportunity to evaluate the medium and long-term effects of these vaccines.

  1. English should be improved. Some wording in Portuguese is present in the manuscript. Eg. Page 7 after table.

EAPV = postvaccination adverse event. Research Source: Brazil's National Immunization Program

  1. References should be confirmed: Eg: "Turkey33", Reference 36 is missing?
  • immunogenicity in healthy adults 18 years of age and older. However, there are reports of pityriasis rosea that developed 4 days after the first dose of the vaccine and were present for 1 week in a phase 3 clinical trial conducted in Turkey33.
  • Corroborating this statement, the reports reported by Carbinatto36 point out that different sequences indicate that SARS-Cov-2 mutates, forming "sub-groups" of the same virus. According to estimates, the study points out, this has been happening at a fre-quency of about one mutation per month. But these changes also reveal which viruses are closest to each other and which are more distant – by allowing the construction of a "family tree".
  •  
  •  
  • Once again, thank you for all your contributions!
    Yours sincerely,
    Mariana Carvalho

Reviewer 3 Report

Comments and Suggestions for Authors

The manuscript presented by the authors is a very interesting review and provides information that will be useful in the field of pharmacovigilance. However, I consider that the reviews could be substantially improved if the authors integrate a quantitative analysis. Considering the following perspective that has been reported in other studies:

M. Aria and C. Cuccurullo, “bibliometrix: An R-tool for comprehensive science mapping analysis,” J. Informetr., vol. 11, no. 4, 684 pp. 959–975, 2017. 685

L. Waltman, N. J. van Eck, and E. C. M. Noyons, “A unified approach to mapping and clustering of bibliometric networks,” J. 686 Informetr., vol. 4, no. 4, pp. 629–635, 2010.

An approach that would allow graphically integrating the relevance of the reviewed articles. I understand that it is a narrative analysis, but some tables with percentage values grouping their results would be very useful for readers.

It would be ideal to adjust the language style.

It would be relevant for the authors to review:

Page, M.J., McKenzie, J.E., Bossuyt, P.M. et al. The PRISMA 2020 statement: an updated guideline for reporting systematic reviews. Syst Rev 10, 89 (2021). https://doi.org/10.1186/s13643-021-01626-4

The introduction requires more support and I am concerned that they only have seven references.

Finally, in the discussion, the authors should make some contrasts with references from related studies.

Comments on the Quality of English Language

Moderate editing of English language required

Author Response

Dear reviewer,

I would like to express my sincere gratitude for the time and effort dedicated to reviewing the scientific article, entitled "PHARMACOVIGILANCE IN BRAZIL: GOVERNMENTAL MONITORING OF ADVERSE SIDE EFFECTS CAUSED BY VACCINES AGAINST COVID-19 – A NARRATIVE REVIEW". His experience and insight were invaluable in improving the quality and clarity of the work.

His detailed observations and constructive suggestions demonstrated a deep knowledge of the subject, significantly enriching the content. I would especially like to thank you for your patience and dedication in pointing out specific aspects that needed attention, which undoubtedly contributed to raising the quality of the article as a whole.

Furthermore, your quick response and availability to provide feedback were essential for meeting the established deadlines. Your attention to detail and commitment to academic excellence have not gone unnoticed and are truly appreciated.

Answer 1: ​The choice of a qualitative analysis is justified by the complexity and multifaceted nature of the data available in the first months of vaccine use. Therefore, the temporal restriction of the study limits the immediate analysis of the acute adverse effects of vaccines against COVID-19. However, it sets the stage for future research, highlighting the opportunity to evaluate the medium and long-term effects of these vaccines.

Answer 2: The flowchart was created to illustrate the data selection process and was not inspired by the Preferred Reporting Items for Systematic Reviews and Meta-Analyses (PRISMA) diagram, as the present study is a narrative review. As PRISMA is an evidence-based minimum set of items for reporting in systematic reviews and meta-analyses, it is not applied to the present study.

Answer 3: The English has been reviewed by the authors to ensure consistency and coherence throughout the document. This includes consistent use of terminology, logical presentation of arguments, and uniformity in formatting.

The text was restructured to provide greater clarity and avoid ambiguities and thus, it was possible to improve the quality of the article, making it more accessible and understandable for the target audience.

Thanks again for all your contributions!

Best regards,

Mariana Carvalho

Reviewer 4 Report

Comments and Suggestions for Authors

The manuscript healthcare-2788140 titled "PHARMACOVIGILANCE IN BRAZIL: THE GOVERNMENT MONITORING OF THE SIDE EFFECTS CAUSED BY VAC-CINES AGAINST COVID-19 – A NARRATIVE REVIEW" aimed to evaluate pharmacovigilance as a resource for controlling and understanding the adverse effects caused by vaccines in use. The authors systematically search relevant articles and narratively review the topic. The authors can improve the paper by addressing the following comments:

1. It is not clear about the periods of relevant articles the authors want to include. Is it May 2021 to June 2022? why do the authors limit in a short period?

2. Table 2. How are the total number of people vaccinated?

3. A list of adverse event and its severity categories is recommended.

4. How can this paper conclude a "proven efficacy" when there is no analysis for that?

5. How can the authors conclude that the adverse events and death rate are "acceptable'? Which evidence do you use?

6. Results section of the abstract is too short compared to other parts of the abstract.

Author Response

Dear Reviewer,

I would like to express my sincere gratitude for the time and effort dedicated to reviewing the scientific article, entitled "PHARMACOVIGILANCE IN BRAZIL: THE GOVERNMENT MONITORING OF THE ADVERSE SIDE EFFECTS CAUSED BY VACCINES AGAINST COVID-19 – A NARRATIVE RE-VIEW". His expertise and insight were invaluable in improving the quality and clarity of the work.

His detailed observations and constructive suggestions demonstrated a deep understanding of the subject, significantly enriching the content. I am especially grateful for your patience and dedication in pointing out specific aspects that required attention, which undoubtedly contributed to raising the quality of the article as a whole.

Furthermore, your quick response and readiness to provide feedback were essential in meeting the established deadlines. Your attention to detail and commitment to academic excellence have not gone unnoticed and are truly appreciated.

he manuscript healthcare-2788140 titled "PHARMACOVIGILANCE IN BRAZIL: THE GOVERNMENT MONITORING OF THE SIDE EFFECTS CAUSED BY VAC-CINES AGAINST COVID-19 – A NARRATIVE REVIEW" aimed to evaluate pharmacovigilance as a resource for controlling and understanding the adverse effects caused by vaccines in use. The authors systematically search relevant articles and narratively review the topic. The authors can improve the paper by addressing the following comments:

  1. It is not clear about the periods of relevant articles the authors want to include. Is it May 2021 to June 2022? why do the authors limit in a short period?

Certainly, it is relevant to discuss the temporal limitation of the study concerning the acute adverse effects of COVID-19 vaccines. The data collection confined between May 2021 and 2022 may constrain the comprehensive analysis of immediate adverse effects of the vaccines, given that the vaccination commenced in December 2020.

However, it is crucial to emphasize that this temporal restriction does not negate the significance of the study. Although the research has a limited scope in assessing the vaccines' acute effects, it can provide a robust foundation for exploring mid- to long-term effects. This limitation, in fact, may pave the way for future longitudinal investigations, enabling a more comprehensive and detailed analysis of potential adverse effects over a broader timeframe.

Thus, while the temporal constraint may limit the immediate examination of acute adverse effects of the vaccines, it sets a promising stage for ongoing research, facilitating broader inquiries into the mid- and long-term impacts of COVID-19 vaccines.

  1. Table 2. How are the total number of people vaccinated?

These data were not presented in the source consulted.

  1. A list of adverse event and its severity categories is recommended.

Unfortunately, the data does not allow us to make a list of severity categories and adverse effects.

  1. How can this paper conclude a "proven efficacy" when there is no analysis for that?

The text has been changed to: The vaccines in use have proven immunogenicity, despite adverse effects that cannot be attributed solely and exclusively to their use, since the medical history of each patient should be considered in the investigation of each case, especially when there is death.

  1. How can the authors conclude that the adverse events and death rate are "acceptable'? Which evidence do you use?

Due to the size of the Brazilian population and vaccination coverage in Brazil, incidents of adverse effects were considered low.

  1. Results section of the abstract is too short compared to other parts of the abstract.

The abstract has been reformulated.

Once again, thank you for all your contributions!

Yours sincerely,

Mariana Carvalho

Round 2

Reviewer 1 Report

Comments and Suggestions for Authors

The authors addressed the majority of the issues raised from introduction to conclusion

Comments on the Quality of English Language

Quality of English is readable

Author Response

Dear Reviewer,

Thank you for your careful review of my article. Your suggestions and corrections were extremely useful in improving the quality of the work. I would like to express my gratitude for the valuable insights you have provided.

Reviewer 2 Report

Comments and Suggestions for Authors

Abstract

Why the monitoring is from the Government? Is not performed by ANVISA?

The objective on the abstract is ambiguous. What is your objective? Evaluate pharmacovigilance as a resource of doing what pharmacovigilance does? Useless redundact objective. Rephrase. The objective is to do a narrative review...

What is "eletronic media"? Rephrase. 

This is a narrative review. The overall number of references is poor. Focusing on Brasil, almost half of references are out of Brasil. Provide more evidence from the Brasil. 

Once again, I add the same question. Covid started at March 2020. Why the search is provided from May 2021 to June 2022. At that point I expect an update of the review from March 2020 to December 2023. 

Abstract results: 

"The occurrence of some adverse events was observed (which?), including cases of allergy (how many?), myocarditis (how many?) and rheumatoid arthritis (how many?). It is important to highlight that these adverse effects were identified as rare, occurring in a small percentage (how many?) of the vaccinated population (which is the incidence?)

Introduction: Rephrase the objetive - this is a narrative review. The objective should be do a narrative review...

In Table 1 - why 4 searches were performed to PubMed but only 3 to others?

Minor:

A lot of words in abstract have hifens on that. Correct. 

Comments on the Quality of English Language

English should be reviewed by a native or proofreading service. Very difficult to understand. 

Author Response

Caro revisor,

Espero que você esteja bem. Gostaria de expressar minha sincera gratidão pelo tempo e esforço que você dedicou à revisão detalhada do meu artigo. Suas críticas construtivas foram fundamentais para obter uma compreensão mais profunda das áreas que precisam de melhorias.

Compreendo a natureza franca das suas observações e quero assegurar-lhe que as recebo com respeito e abertura. É através de feedback rigoroso como o seu que podemos melhorar a qualidade e o impacto do nosso trabalho.

1- Resposta: A Agência Nacional de Vigilância Sanitária (ANVISA) é um órgão federal brasileiro, vinculado ao Ministério da Saúde, responsável pela regulação e monitoramento de produtos e serviços que impactam a saúde da população. Criada em 1999 pela Lei nº 9.782, a ANVISA desempenha papel crucial na promoção e controle da saúde no Brasil.

2- Resposta: Sim, o objetivo do estudo foi avaliar a farmacovigilância como recurso para controle e compreensão dos efeitos colaterais adversos causados ​​pelas vacinas em uso.
A farmacovigilância é um recurso essencial para monitorar a segurança de medicamentos e outros produtos de saúde após sua comercialização e uso em larga escala. Seu principal objetivo é detectar, avaliar, compreender e prevenir eventos adversos ou qualquer problema relacionado ao uso de medicamentos, contribuindo para a promoção da segurança do paciente.

No entanto, a abordagem não é redundante porque durante o período pandémico a farmacovigilância tornou-se ainda mais crucial devido a vários factores:

Rápido Desenvolvimento de Vacinas: O contexto da pandemia exigiu o desenvolvimento acelerado de medicamentos e vacinas. A farmacovigilância tem desempenhado um papel fundamental na avaliação contínua da segurança destes produtos, fornecendo dados sobre potenciais efeitos secundários ou preocupações de segurança.

Uso emergencial de vacinas: Em muitos casos, os medicamentos foram autorizados para uso emergencial. A farmacovigilância foi fundamental para monitorar esses medicamentos em tempo real, identificando rapidamente qualquer sinal de risco ou evento adverso.

Perturbações nos sistemas de saúde: O sistema de saúde foi intensamente afetado pela pandemia, com hospitais e profissionais de saúde sobrecarregados. A farmacovigilância ajudou a manter a vigilância sobre a segurança dos medicamentos, apesar das pressões sobre o sistema de saúde.

Vacinação em massa: Com as campanhas de vacinação em massa, a farmacovigilância tem sido vital para monitorizar a segurança das vacinas contra a COVID-19, fornecendo informações valiosas sobre eventos adversos raros, interacções medicamentosas e outras questões relacionadas com a imunização em grande escala.

A revisão narrativa foi o método utilizado no estudo para atingir o objetivo proposto.

3- Resposta: o termo foi reformulado para bases de dados.

4- Resposta: Obrigado pelo seu valioso feedback sobre nossa revisão narrativa. Reconhecemos a importância de fornecer uma base bibliográfica mais abrangente, especialmente com foco mais específico no contexto brasileiro.

Compreendemos a observação sobre o número limitado de referências e a presença significativa de fontes fora do Brasil. Reconhecemos que uma abordagem mais equilibrada, com maior representação de estudos e dados nacionais, fortalecerá a qualidade e a relevância do trabalho.

É importante destacar que, por decisão dos autores, o período da pesquisa foi mantido, limitando a inclusão de dados mais recentes. Entendemos que esta escolha pode ter impactado no alcance dos resultados. Contudo, gostaríamos de assegurar que estamos empenhados em melhorar o nosso trabalho e futuramente apresentaremos um novo artigo com dados mais robustos e atuais. Esta será uma prioridade para nós, visando enriquecer a revisão narrativa com informações mais recentes e específicas do contexto brasileiro.

Obrigado novamente por sua contribuição construtiva e crítica.

Reviewer 3 Report

Comments and Suggestions for Authors

Dear authors,

Very kind for making the indicated adjustments and corrections, so the manuscript improved substantially and only requires a fine revision of language style.

Kind regards,

Author Response

Caro revisor,

Obrigado por sua revisão cuidadosa do meu artigo. Suas sugestões e correções foram extremamente úteis para melhorar a qualidade do trabalho. Gostaria de expressar minha gratidão pelos valiosos insights que você forneceu.

Reviewer 4 Report

Comments and Suggestions for Authors

The authors completely addressed my comments.

Author Response

Caro revisora,

Obrigado por sua revisão cuidadosa do meu artigo. Suas sugestões e sugestões foram extremamente úteis para melhorar a qualidade do trabalho. Gostaria de expressar minha gratidão pelos insights importantes que você veio.